# Exposure of Urban European Hedgehogs (*Erinaceus europaeus*) to *Toxoplasma gondii* in Highly Populated Areas of Northeast Spain

**DOI:** 10.3390/ani14111596

**Published:** 2024-05-28

**Authors:** Alejandra Escudero, Maria Puig Ribas, Elena Obón, Sonia Almería, Xavier Fernández Aguilar, Hojjat Gholipour, Oscar Cabezón, Rafael Molina-López

**Affiliations:** 1Anatomía Patológica, Departamento de Producción y Sanidad Animal, Facultad de Veterinaria, Universidad Cardenal Herrera-CEU, 46115 Alfara del Patriarca, Valencian Community, Spain; alejandra.escudero@uchceu.es; 2Wildlife Conservation Medicine Research Group (WildCoM), Departament de Medicina i Cirurgia Animals, Universitat Autònoma de Barcelona, 08193 Bellaterra, Catalonia, Spain; mariapuigribas@gmail.com (M.P.R.); gholipour.hojjat@gmail.com (H.G.); 3Centre de Fauna Salvatge de Torreferrussa, Forestal Catalana, S.A., Generalitat de Catalunya, 08130 Santa Perpètua de la Mogoda, Catalonia, Spain; elena.obon@gencat.cat (E.O.); rafael.molina@gencat.cat (R.M.-L.); 4Center for Food Safety and Nutrition (CFSAN), Department of Health and Human Services, Food and Drug Administration, Office of Applied Research and Safety Assessment (OARSA), Division of Virulence Assessment, Laurel, MD 20708, USA; maria.almeria@fda.hhs.gov; 5Unitat Mixta d’Investigació IRTA-UAB, Centre de Recerca en Sanitat Animal (CReSA), Campus de la Universitat Autònoma de Barcelona (UAB), 08193 Bellaterra, Catalonia, Spain; 6Department of Clinical Sciences, Faculty of Veterinary Medicine, Shahid Bahonar University of Kerman, Kerman 76179-14111, Iran

**Keywords:** *Toxoplasma gondii*, urban, hedgehog, *Erinaceus europaeus*

## Abstract

**Simple Summary:**

*Toxoplasma gondii* is a generalist zoonotic parasite that involves warm-blooded animals as intermediate hosts and felines as definitive hosts. Recent studies have shown significant positive associations between human population density and *T. gondii* seroprevalence in wildlife. However, there is a lack of data regarding the exposure of *T. gondii* in urban wildlife. The present study aimed to analyse the *T. gondii* exposure of urban hedgehogs from the Metropolitan Area of Barcelona, NE Spain. Our results reported a high seroprevalence of urban hedgehogs to the parasite, reinforcing the association between human population density and environmental *T. gondii* oocysts. Urban hedgehogs could be good candidates for sentinels of the presence of *T. gondii* oocysts in anthropised areas. In addition, the role of vertical transmission of *T. gondii* in hedgehogs, as well as the impact of *T. gondii* on the health of urban wildlife species, needs further research.

**Abstract:**

*Toxoplasma gondii* is a generalist zoonotic parasite that involves a wide range of warm-blooded animals as intermediate hosts and felines as definitive hosts. Recent studies have proved significant positive associations between human population density and *T. gondii* seroprevalence in wildlife. However, there is limited data regarding *T. gondii* wildlife in urban areas, where the highest human density occurs. The present study aimed to analyse the *T. gondii* exposure in urban hedgehogs from the Metropolitan Area of Barcelona, NE Spain. One hundred eighteen hedgehogs were analysed for the presence of antibodies (modified agglutination test; n = 55) and parasite DNA (qPCR; heart = 34; brain = 60). Antibodies were detected in 69.09% of hedgehogs. *T. gondii* DNA was not detected in any of the analysed samples. The present study reports a high *T. gondii* seroprevalence in urban hedgehogs in areas surrounding Barcelona, the most densely human-populated area of NE Spain, reinforcing the association between human population density and environmental *T. gondii* oocysts. The lack of detection by molecular techniques warrants more studies. In the last few decades, the distribution and abundance of European hedgehogs have declined, including their urban populations. Further research is needed to investigate the impact of *T. gondii* on hedgehog populations.

## 1. Introduction

*Toxoplasma gondii* is a generalist zoonotic parasite with a complex life cycle that involves warm-blooded animals as intermediate hosts. Domestic and wild felid species play a critical role in the ecology and epidemiology of *T. gondii* because they are the definitive hosts, excreting oocysts in their faeces. Toxoplasmosis has been reported worldwide, causing mild disease in humans, domestic animals, and wildlife; however, it can be fatal in young, immunocompromised, pregnant, or congenitally infected individuals [1]. The epidemiology of *T. gondii* is influenced by several factors such as parasite genotype, wild/domestic feline population characteristics, hosts’ interactions, environment, or anthropogenic factors including land use, environmental degradation, and climate change [1].

The transmission of zoonotic infectious agents is widely related to the interactions between wild and domestic animals and humans. In this sense, urban areas are potential high-risk spots for pathogen transmission [2,3,4,5]. In urban areas, feral cat populations reach high densities, increasing the probability of *T. gondii* transmission between hosts [5]. From this perspective, recent studies have shown significant positive associations between human population density and *T. gondii* oocysts shedding in domestic and wild felids [5] and seroprevalence in wild mammals [4,6,7]. Therefore, urbanisation may create ideal environmental conditions for the ecology of this parasite [5,8,9].

Although *T. gondii* has been extensively described in wild fauna from Spain [10,11,12,13,14,15] and also in the definitive host in urban areas of Barcelona [16], little data have been reported from urban wildlife. Hence, the analysis of the exposure of urban wildlife to *T. gondii* is important to increase the knowledge of its epidemiology in anthropised areas. The European hedgehog (*Erinaceus europaeus*) is an omnivorous nocturnal animal distributed throughout the Mediterranean basin, including urban areas [17]. They typically live in anthropogenic areas and are susceptible to infection by *T. gondii* [18]. Due to its adaptability to anthropised habitats, this species could be considered a suitable sentinel for the environmental presence of zoonoses in urban areas. The aim of the present study was to analyse the prevalence of *T. gondii* infection in free-ranging European hedgehogs from urban ecosystems in the most densely human-populated area of NE Spain.

## 2. Materials and Methods

### 2.1. Animals and Samples

One hundred eighteen diseased European hedgehogs euthanised for welfare reasons at the Wildlife Rehabilitation Centre (WRC) of Torreferrussa in Barcelona (Forestal Catalana, Catalonia Government, license number B2300083), Catalonia (NE Spain) were analysed. The animals studied were collected within the Metropolitan Area of Barcelona (MAB), the most densely human-populated area of NE Spain [19] (Figure 1). Animals were classified as sub-adults (<1 year old) and adults (life expectancy is 3 years (0–10 years) [17]). The sex of the animals was recorded. Sera, heart, and brain were analysed, taking advantage of the biological samples archive of the WRC of Torreferrussa: 55 animals (sera), 29 animals (brain); 31 animals (brain + heart), 3 animals (heart).

### 2.2. Laboratorial Analyses

Sera were examined by the modified agglutination test (MAT) to detect antibodies against *T. gondii* as previously described [20]. Each serum sample was tested at dilutions of 1:25, 1:50, and 1:500. Previously, IgM antibodies from sera were neutralised using 2-mercaptoethanol. Titres of 1:25 or higher were considered positive. DNA was extracted from the brain (0.2 g) and heart (0.2 g) tissues using the commercial kit NucleoSpin Tissue (Macherey-Nagel, Düren, Germany) according to the manufacturer’s procedure. Extracted DNA was amplified using a real-time PCR (rtPCR) with Toxo-SE (5′ AGGCGAGGGTGAGGATGA 3′) and Toxo-AS (5′ TCGTCTCGTCTGGATCGCAT 3′) primers, and the probe (5′ 6FAM-CGACGAGAGTCGGAGAGGGAGAAGATGT-BHQ1 3′), using a commercial kit (TaqMan PCR Master Mix; Applied Biosystems, Carlsbad, CA, USA) [21]. Primers Toxo-SE and Toxo-SA target the 529 bp repeat region (REP529, GenBank accession no. AF146527) of *T. gondii*. The qPCR method used can detect *T. gondii* DNA extracted from a single cyst DNA [21,22]. DNA was extracted from *T. gondii* oocysts (purchased at Grupo SALUVET, Departamento de Sanidad Animal, Facultad de Veterinaria, Universidad Complutense de Madrid, Spain) and used as DNA positive control. Each qPCR run included a negative control, containing 500 µL of PBS. The cycling protocol was as follows: 50 °C for 2 min (activation of the uracil-N-glycosylase) and denaturation at 95 °C for 10 min, followed by 40 cycles at 95 °C for 15 s and 61 °C for 1 min.

### 2.3. Statistical Analyses

Differences in *T. gondii* seroprevalence between age groups (subadults and adults) and sex (males and females) were assessed using Pearson’s chi-squared test, with statistical significance determined at *p* < 0.05. Seroprevalence was estimated using the 95% Wilson confidence interval. All statistical analysis were performed using the R 3.1.2 GUI 1.65 Statistical Program and the “epiR” package to calculate prevalence estimates.

## 3. Results

Antibodies against *T. gondii* were detected in 38 out of 55 (69.09%; 95%CI: 56.0–79.7) hedgehogs. *Toxoplasma gondii* DNA was not detected in any of the 34 hearts (0.0%; 95%CI: 0.0–10.1) and 60 brains (0.0%; 95%CI: 0.0–6.0) analysed. No statistically significant differences were found in *T. gondii* seroprevalence between age groups (sub-adults, 64.52%; adults, 75%) (X^2^ = 0.69, *p* = 0.40) or sex (females, 74.95%; males, 64.28%) (X^2^ = 0.62, *p* = 0.43).

## 4. Discussion

Landscape anthropization creates the perfect environmental conditions for some domestic animals, favouring the overabundance of some of these species, including domestic cats, the definitive host of *T. gondii*, and their related pathogens [23], with adverse effects on wild species, including urban wildlife. Understanding the dynamics of *T. gondii* in urban areas requires the knowledge of many factors (i.e., local climate, land use, habitat composition, domestic/wild felines’ abundance, intermediate hosts community composition) that may influence its circulation at a local scale. The present study provides information on *T. gondii* seroprevalence of urban hedgehogs from the most densely human-populated area of NE Spain, the Metropolitan Area of Barcelona.

Mammals from a high trophic level, omnivorous or carnivorous species, can be good sentinels for the detection of pathogens in certain ecosystems [15,24]. Omnivorous and carnivorous animals have been increasingly drawn to urban or suburban areas due to easy access to food or shelter [25]. Previous studies have used such high trophic level species, such as American minks as sentinels or indicators for the circulation of *T. gondii* in semiaquatic freshwater ecosystems [15,26], sea otters in marine environments [27] and hedgehogs as sentinels in urban areas [18]. Our results agree with the previous report on hedgehogs in urban areas in the Czech Republic [18] and suggest that hedgehogs can be used as an indicator for the presence of *T. gondii* circulating in areas located near human settlements.

A high seroprevalence of antibodies against *T. gondii* was observed in the studied urban hedgehogs in the MAB. Previous serological studies in hedgehogs showed lower seroprevalence in Austria (25% of 64 hedgehogs by complement fixation test (CFT)) [28] and in central Italy (19% of 100 hedgehogs by Dye test) [29]. In 40 hedgehogs tested serologically in Teramo Province, Italy, no positivity was found [30]. The high prevalence of antibodies against *T. gondii* assessed in the studied urban hedgehogs is similar to the high antibody prevalences found in feral cats (51.9%) [16] and pregnant women (28.6%) [31] in the same MAB. Similarly, significantly higher seroprevalence levels have been reported in wildlife from the same region, in Catalonia, NE Spain, compared to other areas of Spain in several species, including wild rabbits (53.8%) [32], red deer (42.2%) [16], wild birds (27.2%) [11], or even in legally trapped and released common ravens in Catalonia, which showed one of the highest seroprevalences in wild birds worldwide (80.5%) [14]. MAT has been widely used and accepted in epidemiological studies of different wildlife species. Although the specificity and sensitivity of MAT have not been evaluated for the diagnosis of toxoplasmosis in the European hedgehog, it is the most sensitive and specific test for the diagnosis of toxoplasmosis in warm-blooded species.

However, several studies have reported differences in *T. gondii* infection in small mammals from different highly anthropised areas in Europe [7,18,33,34,35,36,37,38,39,40,41] (Table 1), suggesting different rates of exposure to the parasite associated with specific factors from the studied areas (i.e., environmental conditions, host communities, host susceptibility, *T. gondii* lineages).

Importantly, when *T. gondii* was studied in the invasive American mink in Catalonia, a high seroprevalence (82.6%; n = 46) was observed [15]. In that study, *T. gondii* was found in American mink brain tissues (9.2%; n = 120) using the same molecular detection method as in the present study, while no *T. gondii* DNA was detected in the analysed hedgehogs. In the Czech Republic, the parasite was detected in 19.2% of 26 European hedgehogs analysed by PCR in brain tissue [18]. A possible explanation for the lack of *T. gondii* DNA detected in the brain and hearts of the hedgehogs analysed in the present study is that the analysis of DNA in tissues has usually lower sensitivity than serology due to the random distribution of tissue cysts, the limited volume of sample analysed (1 g of the sample from the whole organ), and the usual low parasite burden observed in the tissues of chronically infected animals [42]. This makes PCR impractical for prevalence studies of *T. gondii* in chronically infected animals.

The high prevalence of *T. gondii* infection in urban hedgehogs from the MAB can be explained by its habitat overlapping with feral cats. Although several studies have reported that domestic cats show lower prevalences of oocyst shedding than wild felines [8,43], the large populations of cats represent a significant source of *T. gondii* oocysts. There are no current estimates of cat densities in the MAB. However, a positive association between human population density and *T. gondii* oocysts shedding prevalence of free-ranging domestic cats has been reported worldwide [5]. The high prevalence of infection found in urban hedgehogs from the MAB reinforces this association and suggests a high load of *T. gondii* oocysts in the environment, representing a health risk for the human and wildlife populations in urban areas, in accordance with previous studies [44,45]. The evidence of accumulating *T. gondii* oocysts in urban areas posing a significant public health hazard has been described worldwide [46].

Many previous studies have shown a relationship of higher seroprevalence by increased age in wildlife species as an indication of continuous contact with the parasite [15,18]. Due to the lifelong persistence of *T. gondii* IgG antibodies in healthy individuals, seroprevalence by MAT reflects lifelong exposure to the parasite. Increased contact by age indicates horizontal transmission as the main route of *T. gondii* infection. However, in the present study, there were no significant differences by age groups (sub-adults, 64.52%; adults, 75%) and these results could imply vertical transmission as an important route of *T. gondii* transmission in hedgehogs in the area, as proved in some small mammals [47]. However, further studies are needed to verify this hypothesis.

In Europe, the expansion of urbanised areas into natural habitats and the increase in urban wildlife in green zones and parks in populated areas favour some wildlife species [48,49]. One of these species is the European hedgehog [50,51,52,53]. However, in the last few decades, the distribution and abundance of European hedgehogs have declined in different European countries, including urban populations [49,54,55,56,57,58,59,60]. Although several causes have been proposed for their decline in urban areas, including parasitic diseases (e.g., *Crenosoma striatum*, *Capillaria* spp., gastropod-transmitted lungworms) [49], further research is necessary to investigate the impact of *T. gondii* on the hedgehogs’ population health.

## 5. Conclusions

The high *T. gondii* seroprevalence observed in hedgehogs in the MAB indicates that urban hedgehogs are good candidates to be sentinels of the presence of *T. gondii* oocysts in anthropised areas. The fact that there were no significant differences in seroprevalence related to age could be an indication of vertical transmission of *T. gondii* in this species, but further research is needed. In addition, further research is also needed about the potential impact of *T. gondii* on the health of urban wildlife species, such as hedgehogs.

## Figures and Tables

**Figure 1 animals-14-01596-f001:**
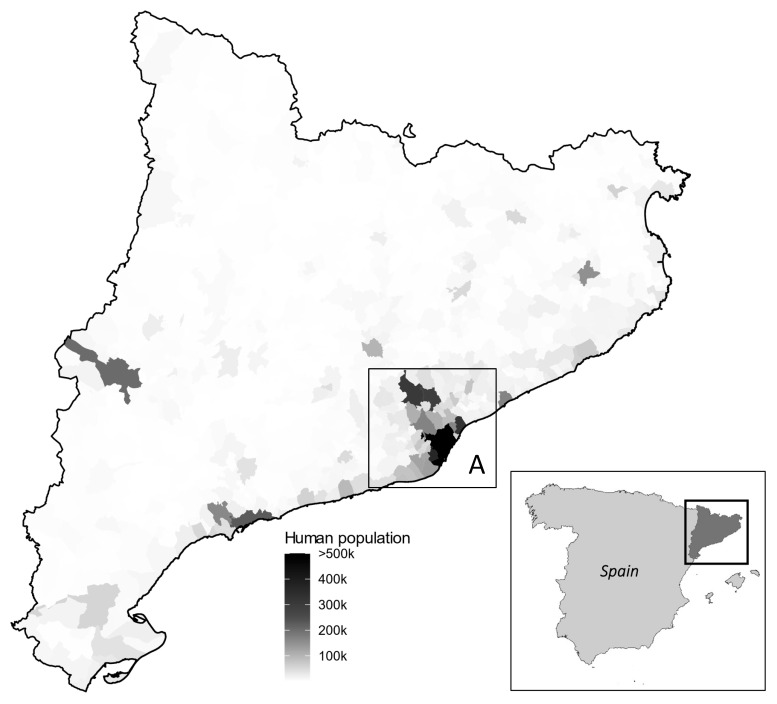
The hedgehogs analysed in this study were found in the Metropolitan Area of Barcelona, the largest human population of NE Spain (A). A gradient of the human population is depicted in the panel.

**Table 1 animals-14-01596-t001:** Prevalence of *Toxoplasma gondii* infection in small mammals from highly anthropised areas from Europe.

Species	% *T. gondii*	Technique	Tissue	Country	Reference
*Mus musculus*	53%	PCR	brain	UK	[33]
*Rattus norvegicus*	42.2%				
Rodents	4.66%	PCR	brain	Berlin (Germany)	[34]
*Apodemus flavicollis*	2.22%				
*Apodemus agrarius*	5.13%				
*Myodes glareolus*	0%				
*Apodemus sylvaticus*	8%				
*Microtus arvalis*	27.3%				
*Microtus agrestis*	50%				
Rodents	2.94%	PCR	brain	Istria (Croatia/Slovenia)	[35]
*Apodemus agrarius*	1.2%	PCR	heart	Romania	[37]
*Apodemus flavicollis*	8.5%				
*Apodemus sylvaticus*	9.1%				
*Micromys minutus*	11.1%				
*Microtus arvalis*	4.3%				
*Microtus subterraneus*	4.4%				
*Mus musculus*	5.7%				
*Myodes glareolus*	35.5%				
*Sorex araneus*	21.7%				
*Sorex minutus*	23.1%				
*Spermophilus citellus*	33.3%				
*T. europaea*, *R. norvegicus*, *O. zibethicus*, *N. anomalus*, *M. spicilegus*, *C. suaveolens*, *C. leucodon*, *A. amphibious*, *A. uralensis*	0%				
*Erinaceus europaeus*	19.2%	PCR	brain	Czech Republic	[18]
*Erinaceus roumanicus*	16.6%				
*Rattus rattus*	26.9%	IFAT	serum	Cyprus	[38]
*Rattus norvegicus*	28.3%				
*Rattus norvegicus*	7.7%	MAT	serum	Rhône (France)	[39]
*Rattus norvegicus*	27.5%	MAT	serum	Serbia	[40]
*Erinaceus europaeus*	19%	Dye test	serum	Central Italy	[29]
*Erinaceus europaeus*	0%	Dye test	serum	Teramo province, Italy	[30]
*Erinaceus europaeus*	25%	CFT	serum	Austria	[28]
*Castor fiber*	54.6%	ELISA	serum	Switzerland	[7]
*Apodemus flavicollis*	2.5%	ELISA	serum	Geneva (Switzerland)	[41]
*Microtus arvalis*	3.1%				
*Arvicola terrestris*	5%				

MAT: modified agglutination test, CFT: complement fixation test, IFAT: indirect immunofluorescent antibody assay (IFAT).

## Data Availability

The original contributions presented in the study are included in the article, further inquiries can be directed to the corresponding author/s.

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
