# Peer review of "Exposure of Urban European Hedgehogs (Erinaceus europaeus) to Toxoplasma gondii in Highly Populated Areas of Northeast Spain"

_animals, 2024, doi:10.3390/ani14111596_

Round 1

Reviewer 1 Report

Comments and Suggestions for Authors

Line 55 - I suggest that it adds the damage in pregnant and suppressed individuals. Support with bibliography.

Line 81 - Please explain "poor health".

Line 84 - I suggest : " The animals studied.

Line 86 - Please refer the average life expectancy of one hedgehogs.

line 95 - ... were re-examined ... How? Which methodology?!

Line 119 - Please refer "sex" first in the "material and methods", point 2.1.

Line 146 /7 - The "accordance " It's not  linear Thethere are other factors. Please rephrase.

Line 177 -  .... the limited volume of sample analyzed,... What volume was used in the study of reference 42? Please discuss this.

Line 179/80 - ...Metropolitan  Area of Barcelona... please replace MAB.

Line 194 - What is MAT?!

Line 205 - ...infectious diseases ... which microorganisms?

Comments on the Quality of English Language

As I already pointed out ,minor editing of English language is required.

Author Response

Line 55 - I suggest that it adds the damage in pregnant and suppressed individuals. Support with bibliography.

Done (lines 55-56)

Line 81 - Please explain "poor health".

The sentence has been rephrased (line 82).

Line 84 - I suggest : " The animals studied.

Accepted. Done (line 85).

Line 86 - Please refer the average life expectancy of one hedgehogs.

 This information has been added: “Animals were classified as sub-adults (<1 year old) and adults (life expectancy is 3 years (0-10 years) [17]).”

line 95 - ... were re-examined ... How? Which methodology?!

I agree with the reviewer that the sentence can be confusing. The serum samples with doubtful lectures of the MAT were re-tested with the same agglutination technique. In order to facilitate the reading of the manuscript, I delete “…and those with doubtful results were re-examined.“ from the sentence.

Line 119 - Please refer "sex" first in the "material and methods", point 2.1.

Done. Line 87.

Line 146 /7 - The "accordance " It's not  linear Thethere are other factors. Please rephrase.

I agree. The sentence has re-written: “The high prevalence of antibodies against T. gondii assessed in the studied urban hedgehogs is similar with the high antibody prevalences found in feral cats (51.9%) [16], and pregnant women (28.6%) [31] in the same MAB.”

Line 177 -  .... the limited volume of sample analyzed,... What volume was used in the study of reference 42? Please discuss this.

Hill et al. (2006) used 0.5–1 g of tissue (heart, tongue, etc.) for DNA extraction and PCR. Although PCR and qPCR are sensitive methods for the detection of T. gondii DNA, these techniques have the limitation of analyzing only a small portion of the whole organ. This, together with the possibility of low parasite cysts and their heterogeneous distribution in the organ, makes this technique not very useful in prevalence studies. The information of the weight (1g of tissue) vs the whole organ has been included in the paragraph in order to help the reader.

Lines 175 – 181: “A possible explanation for the lack of T. gondii DNA detected in the brain and hearts of the hedgehogs analysed in the present study is that the analysis of DNA in tissues has usually lower sensitivity than serology due to the random distribution of tissue cysts, the limited volume of sample analyzed (1g of sample from the whole organ), and the usual low parasite burden observed in the tissues of chronically infected animals [42]. This makes PCR impractical for prevalence studies of T. gondii in chronically infected animals.”.

Line 179/80 - ...Metropolitan Area of Barcelona... please replace MAB.

Done.

Line 194 - What is MAT?!

Line 95 (M&M section; 2.2.): modified agglutination test (MAT).

Line 205 - ...infectious diseases ... which microorganisms?

The sentence has been rewritten.

Lines 207-211. “Although several causes have been proposed for their decline in urban areas, including parasitic diseases (e.g. Crenosoma striatum, Capillaria spp., gastropod-transmitted lungworms) [49], further research is necessary to investigate the impact of T. gondii on the hedgehogs’ population health.

Reviewer 2 Report

Comments and Suggestions for Authors

In line 40 I think you should use MAT( modified agglutination test) only after you explain the abbreviation.

Just check one more time the text for small typos.

Author Response

In line 40 I think you should use MAT ( modified agglutination test) only after you explain the abbreviation.

This sentence has been rewritten in the abstract (line 40).

Just check one more time the text for small typos.

Thank you for your appreciation. The text has been reviewed.

Reviewer 3 Report

Comments and Suggestions for Authors

Dear authors, this is an interesting study.

You have to explain how many sera samples had paired tissue samples and which ones (brain or heart), and how many heart and brain samples where the only one sample collected from an animal.

Explain why you did not collect from the same hedgehog serum, heart and brain samples.

Author Response

Dear authors, this is an interesting study.

You have to explain how many sera samples had paired tissue samples and which ones (brain or heart), and how many heart and brain samples where the only one sample collected from an animal.

The paragraph has improved: “Sera, heart and brain were analyzed taking of advantage of the biological samples archive of the WRC of Torreferrussa: 55 animals (sera), 29 animals (brain); 31 animals (brain + heart), 3 animals (heart).”.

Explain why you did not collect from the same hedgehog serum, heart and brain samples.

The present study took advantage of the biological samples archive of the Wildlife Rehabilitation Center of Torreferrussa. The sera/tissues analyzed were the ones available in the mentioned archive. The reason why some samples are not analyzed in the present study is because these samples were used in other research studies.

Reviewer 4 Report

Comments and Suggestions for Authors

Interesting study on the occurrence of T. gondii exposure in hedgehogs in an urban area of Spain.

Overall the paper is well presented and written.

I have however, one comment that I think should be addressed: The serological method used for detecting anti T. gondii antibodies is the Modified Agglutination Test (MAT). The reference used for this test in the paper is a report on toxoplasmosis in horses. The MAT can be used on serum samples from humans and different animal species, including domestic animals and wildlife, because it does not use a species-specific conjugate. 

However, the authors should provide information on:

- is it an in house or commercial test? In case of a commercial test, reference?

- Was a 2-mercaptoethanol treatment of serum samples done before using them in the test? It is known that IgM are causing false positive reactions in MAT, therefore the need to eliminate them from the sample before MAT testing.

- How many dilutions were tested in the MAT for each serum sample? How were titres of the positive samples distributed?

- has MAT been used previously in hedgehogs?

- in the M&M section the authors mention a cut off of 1:25. Why having chosen this cut off? They also write that when doubtful, the test was repeated on the same sample. These results should be presented in the results section.

- In the discussion section the validity of the MAT results should be discussed.

In addition, the authors should mention that hedgehogs are omnivorous animals.

Author Response

Interesting study on the occurrence of T. gondii exposure in hedgehogs in an urban area of Spain. Overall the paper is well presented and written.

I have however, one comment that I think should be addressed: The serological method used for detecting anti T. gondii antibodies is the Modified Agglutination Test (MAT). The reference used for this test in the paper is a report on toxoplasmosis in horses. The MAT can be used on serum samples from humans and different animal species, including domestic animals and wildlife, because it does not use a species-specific conjugate. However, the authors should provide information on:

- is it an in house or commercial test? In case of a commercial test, reference?

- Was a 2-mercaptoethanol treatment of serum samples done before using them in the test? It is known that IgM are causing false positive reactions in MAT, therefore the need to eliminate them from the sample before MAT testing.

- How many dilutions were tested in the MAT for each serum sample? How were titres of the positive samples distributed?

- has MAT been used previously in hedgehogs?

- in the M&M section the authors mention a cut off of 1:25. Why having chosen this cut off? They also write that when doubtful, the test was repeated on the same sample. These results should be presented in the results section.

- In the discussion section the validity of the MAT results should be discussed.

I agree with the reviewer’s comments. The reference to Dubey & Desmonts (1987) is due because it was the first research paper where this serological technique was described. But, as the reviewer has pointed out, this technique (homemade) has been used and accepted in a wide variety of warm-blooded animals. The treatment of sera with 2-mercaptoetahnol in order to neutralize the IgM was performed before the agglutination, and the dilutions performed were 1:25, 1:50, 1:100 and 1:500. This information has been included in the M&M section. Any sample presented doubtful results nor showed 1:500 positivity (we would re-analyze at higher dilutions). The dilution 1:25 has been accepted in all previous studies as the cut-off for mammals from many orders, including hedgehogs. The MAT has not been used previously in hedgehogs; this point has been included in the Discussion section (Lines 157-160).

In addition, the authors should mention that hedgehogs are omnivorous animals.

Line 73-74 “The European hedgehog (Erinaceus europaeus) is an omnivorous nocturnal animal distributed throughout the Mediterranean basin, including urban areas [17].”